# RELO: Reinforcement Learning to Localize for Visual Object Tracking

Xin Chen [1]  Chuanyu Sun [1]  Jiao Xu [2]  Houwen Peng [3]  Dong Wang [2]  Huchuan Lu [2]  Kede Ma [1]

https://github.com/Multimedia-Analytics-Laboratory/RELO

## Abstract

Conventional visual object trackers localize targets using handcrafted spatial priors, often in the form of heatmaps. Such priors provide only surrogate supervision and are poorly aligned with tracking optimization and evaluation metrics, such as intersection over union (IoU) and area under the success curve (AUC). Here, we introduce RELO, a REinforcement-learning-to-LOcalize method for visual object tracking that formulates target localization as a Markov decision process. Specifically, RELO replaces handcrafted spatial priors with a localization policy learned over spatial positions via reinforcement learning, with rewards combining frame-level IoU and sequence-level AUC. We additionally introduce layer-aligned temporal token propagation to improve semantic consistency across frames, with negligible computational overhead. Across multiple benchmarks, RELO achieves superior results, attaining $57.5\%$ AUC on LaSOT$_{\text{ext}}$ without template updates. This confirms that reward-driven localization provides an effective alternative to prior-driven localization for visual object tracking.

## 1. Introduction

Visual object tracking aims to estimate the state (mainly the location) of a target throughout a video given its initialization in the first frame (Wu et al., 2013). As a fundamental problem in computer vision, it underpins a wide range of applications, including surveillance, autonomous systems, robotics, and human-computer interaction. Recent progress has been driven by discriminative tracking paradigms, especially Siamese-based methods (Li et al., 2018; Bhat et al., 2019; Chen et al., 2021; Wang et al., 2021a) and one-stream

[1]City University of Hong Kong [2]Dalian University of Technology [3]Hunyuan Team, Tencent. Correspondence to: Kede Ma <kede.ma@cityu.edu.hk>.

*Proceedings of the 43$^{rd}$ International Conference on Machine Learning*, Seoul, South Korea. PMLR 306, 2026. Copyright 2026 by the author(s).

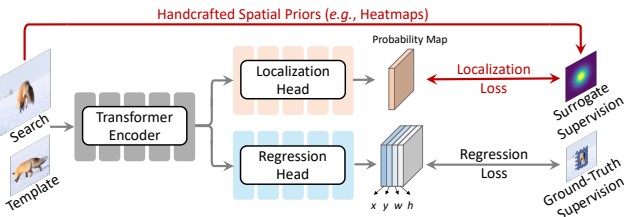

*(a)* Prior-driven localization learning

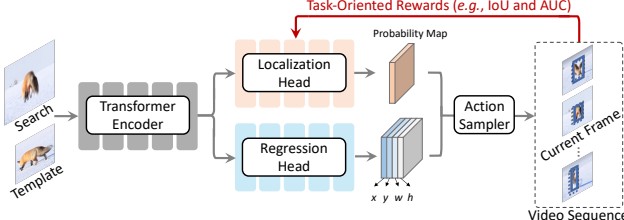

*(b)* RL-based localization learning

*Figure 1.* Comparison of target localization learning paradigms in visual object tracking. **(a)** Conventional trackers learn target localization from handcrafted spatial priors. **(b)** RELO replaces handcrafted priors with an RL-based localization policy, optimized using task-oriented rewards. Red arrows indicate the signals used to optimize the localization module.

Transformer-based trackers (Ye et al., 2022; Zheng et al., 2024; Chen et al., 2025; Lin et al., 2024), which localize the target by learning dense spatial representations over the search region. Despite their strong empirical performance, most of these trackers still rely on handcrafted spatial priors for surrogate supervision, typically in the form of center-based heatmaps that represent target likelihood (see Fig. 1(a)). Several earlier Transformer-based approaches (Cui et al., 2022; Song et al., 2022; Chen et al., 2022; Gao et al., 2022) follow a similar idea in a less explicit manner, by using expectation-based corner localization.

Although effective in practice, such prior-driven localization strategies have an important conceptual limitation: the supervisory signal is only indirectly related to the actual goal of tracking. Handcrafted center or corner distributions encourage the tracker to match a manually specified spatial pattern, rather than directly improving the metrics used to evaluate tracking performance, such as intersection over union (IoU)

and area under the success curve (AUC). In addition, the design of these priors introduces extra assumptions that are not intrinsic to tracking. For example, some trackers adopt binary masks (Chen et al., 2021) and Gaussian-smoothed heatmaps (Ye et al., 2022), and the resulting performance can depend on design choices that are largely heuristic.

These observations motivate a different question: can a tracker learn *where* to localize the target directly from task-oriented feedback, without relying on handcrafted spatial priors? In this paper, we answer this question affirmatively by introducing **RELO**, a **RE**inforcement-learning-to-**LO**calize method for target localization in visual tracking. As shown in Fig. 1(b), RELO formulates target localization as a Markov decision process over the spatial feature map. Each spatial location is treated as a candidate action, the regression head predicts a corresponding bounding box for every action, and a localization policy learns to select promising actions. Instead of fitting the localization module to surrogate labels, RELO optimizes the policy directly using task-oriented rewards that combine frame-level IoU and sequence-level AUC. In this way, localization behavior emerges from the tracking objective itself, rather than from manually designed supervision.

RELO offers two key advantages. First, it aligns training more closely with evaluation by directly optimizing rewards that reflect tracking quality. Second, it removes the need for handcrafted spatial priors, allowing the tracker to discover effective localization behavior through reward-driven exploration. To make this reinforcement learning (RL) process stable and effective, we introduce a regression warmup stage that equips the tracker with basic box regression capability before policy optimization. We additionally incorporate a layer-aligned temporal token propagation strategy to improve semantic consistency across frames with negligible computational overhead.

Extensive experiments on multiple benchmarks demonstrate that RELO achieves state-of-the-art performance on large-scale tracking benchmarks while maintaining real-time inference speed on GPU. In particular, RELO-L256 attains 75.1% AUC on LaSOT (Fan et al., 2019) without template updates and 57.5% AUC on the more challenging LaSOT$_{ext}$ benchmark (Fan et al., 2021), showing strong generalization.

In summary, our contributions are as follows:

- We reformulate target localization in visual tracking as a Markov decision process and instantiate this formulation in RELO, without relying on handcrafted spatial priors.
- We introduce a practical training strategy for RELO, including a regression warmup stage and a layer-aligned temporal token propagation mechanism.
- We show that RELO achieves strong accuracy and efficiency, validating that directly optimizing task-oriented

rewards offers a strong alternative to conventional prior-driven localization.

## 2. Related Work

In this section, we position RELO with respect to the two closely related lines of research: dense-prediction-based and RL-based visual tracking.

### 2.1. Dense-Prediction-Based Visual Tracking

Modern visual object trackers are largely built on discriminative dense prediction, with one-stream Transformer-based trackers emerging as a particularly strong and simple instantiation. These methods typically extract spatial representations using deep visual backbones (He et al., 2022; Oquab et al., 2024; Zhang et al., 2023) and predict the target state from the resulting *search-region* feature map. However, a common characteristic is that target localization is usually learned through handcrafted spatial priors rather than directly end-to-end optimized.

**Center-based localization.** Most recent trackers fall into this category (Xu et al., 2020; Xie et al., 2022; Tang & Ling, 2022; Mayer et al., 2022; Lin et al., 2022; Hong et al., 2024; Zheng et al., 2025; Cai et al., 2025; Zhang et al., 2025). They typically combine a classification-style localization branch, which estimates the confidence of target presence at each spatial location, with a regression branch that predicts the final bounding box. The localization branch is supervised by a center-based prior, usually implemented as either a binary target map (Chen et al., 2021; Xu et al., 2020) or a Gaussian-smoothed heatmap centered at the target position (Ye et al., 2022; Lin et al., 2024).

**Corner-based localization.** Several earlier trackers instead adopt corner-based formulations (Cui et al., 2022; Song et al., 2022; Chen et al., 2022; Gao et al., 2022). These methods predict probability distributions for the top-left and bottom-right corners, and recover the bounding box by taking expectations over the corresponding spatial distributions. Although this avoids an explicit center heatmap, it still introduces handcrafted geometric bias, since the supervisory signal is defined around annotated corners rather than directly around tracking performance.

In contrast, RELO formulates target localization as a Markov decision process and optimizes it with RL using task-oriented rewards, thereby removing manually designed spatial priors and aligning training more closely with testing.

### 2.2. RL-Based Visual Tracking

RL (Sutton & Barto, 1998) has been explored in visual object tracking from several perspectives, but rarely as a

mechanism that directly drives target localization. Early RL-based trackers (Yun et al., 2017; Ren et al., 2018) treat tracking as iterative bounding box refinement, where an agent selects discrete translation or scale actions to adjust a predicted box, typically under coarse IoU-thresholded rewards. Later work extends this idea with continuous actions, hierarchical policies, or teacher-guided distillation (Dunnhofer et al., 2021; Zhang et al., 2021a), yet RL is still primarily used for post-hoc box adjustment.

Beyond bounding box refinement, RL has also been applied to auxiliary online decisions, including active tracking via camera control (Luo et al., 2018), template updating (Zhao et al., 2020; Sun et al., 2020), and adaptive feature or module selection during inference (Zhang et al., 2020; Xie et al., 2018; Huang et al., 2017). More recently, SLT (Kim et al., 2022) uses RL to maximize mean IoU over video clips. However, it is built on top of a strong prior-driven Siamese tracker that provides both initialization and a reference during optimization, so the learned policy remains strongly coupled to the inductive biases of the underlying tracker.

In contrast, RELO applies RL directly to the core task of visual tracking—target localization—by modeling spatial positions on the feature map as actions and learning a localization policy from task-oriented rewards. From this perspective, RL is not an auxiliary control module, but the fundamental mechanism by which localization is learned.

# 3. Proposed Method: RELO

In this section, we present RELO, a visual object tracking method that formulates target localization as a Markov decision process over the spatial feature map of the search region. As shown in Fig. 2, given a template and a search frame, the encoder extracts search features, the regression head predicts four normalized box coordinates at each spatial location, the policy head outputs one logit per location to define the action distribution, and the value head predicts a scalar reward estimate per location.

## 3.1. Problem Formulation

Let $\mathbf{I}_{\text{temp}}$ denote the template image and let $\mathbf{I}^{(t)}$ denote the search image at frame $t$. A one-stream Transformer-based encoder jointly processes the template and search inputs and produces a search feature map $\mathbf{F}^{(t)} \in \mathbb{R}^{H \times W \times C}$, where $H$, $W$, and $C$ denote the height, width, and number of channels, respectively. Each spatial position $(i, j)$ corresponds to a candidate action $a_{ij}$ in the action space $\mathcal{A} = \{(i, j) \mid i \in \{1, \ldots, H\}, j \in \{1, \ldots, W\}\}$. For each action, the regression head predicts a candidate box

$$\mathbf{B}^{(t)} = \boldsymbol{g}_{\text{reg}}\left(\mathbf{F}^{(t)}\right) \in \mathbb{R}^{H \times W \times 4}, \quad (1)$$

where $\boldsymbol{b}_{ij}^{(t)} \in \mathbb{R}^{4 \times 1}$ denotes the box for location $(i, j)$.

During training, we sample a clip of $T$ search frames $\{\mathbf{I}^{(t)}\}_{t=1}^{T}$. At frame $t$, the feature map $\mathbf{F}^{(t)}$ serves as the state representation, and the policy head $\boldsymbol{g}_{\text{policy}}(\cdot)$ induces a categorical distribution over the spatial actions:

$$\pi\left(a_{ij} \mid \mathbf{F}^{(t)}\right) = \frac{\exp\left(m_{ij}^{(t)}\right)}{\sum_{u=1}^{H} \sum_{v=1}^{W} \exp\left(m_{uv}^{(t)}\right)}, \quad (2)$$

where $\mathbf{M}^{(t)} = \boldsymbol{g}_{\text{policy}}(\mathbf{F}^{(t)}) \in \mathbb{R}^{H \times W}$ denotes the logit map at frame $t$. An action $a^{(t)} \sim \pi(\cdot \mid \mathbf{F}^{(t)})$ is sampled during training, and the selected prediction is $\boldsymbol{b}_{a^{(t)}}^{(t)}$. We maximize the expected reward over the sampled clip:

$$J(\pi) = \frac{1}{T} \sum_{t=1}^{T} \mathbb{E}_{a^{(t)} \sim \pi(\cdot \mid \mathbf{F}^{(t)})}\left[r^{(t)}\right], \quad (3)$$

where the reward $r^{(t)}$ is determined by the discrepancy between the predicted box $\boldsymbol{b}_{a^{(t)}}^{(t)}$ and the ground-truth box $\boldsymbol{b}_{\text{gt}}^{(t)}$, with a higher reward assigned to more accurate localization. This formulation removes the need for handcrafted spatial priors; instead, the tracker learns localization behavior directly from task-oriented feedback.

## 3.2. Localization Policy Optimization

We optimize the localization policy using an actor-critic method with a learned value baseline (Sutton & Barto, 1998). For each frame $t$, the value head predicts

$$v^{(t)} = \boldsymbol{g}_{\text{value}}\left(\mathbf{F}^{(t)}\right), \quad (4)$$

which estimates the expected reward conditioned on the current state. The corresponding advantage is

$$A^{(t)} = r^{(t)} - v^{(t)}. \quad (5)$$

Our goal is to maximize the expected reward over the video clip in Eq. (3). In practice, the expectation over actions is approximated by Monte Carlo sampling, *i.e.*, one action is sampled from the policy at each frame. We therefore optimize the following surrogate policy loss:

$$\ell_{\text{policy}} = -\frac{1}{T} \sum_{t=1}^{T} A^{(t)} \log \pi\left(a^{(t)} \mid \mathbf{F}^{(t)}\right), \quad (6)$$

where we replace the raw reward in the policy update with the advantage defined in Eq. (5). The value head is trained by regressing the predicted value to the observed reward:

$$\ell_{\text{value}} = \frac{1}{T} \sum_{t=1}^{T} \left(v^{(t)} - r^{(t)}\right)^2. \quad (7)$$

The final optimization loss is

$$\ell = \ell_{\text{policy}} + \beta \, \ell_{\text{value}}, \quad (8)$$

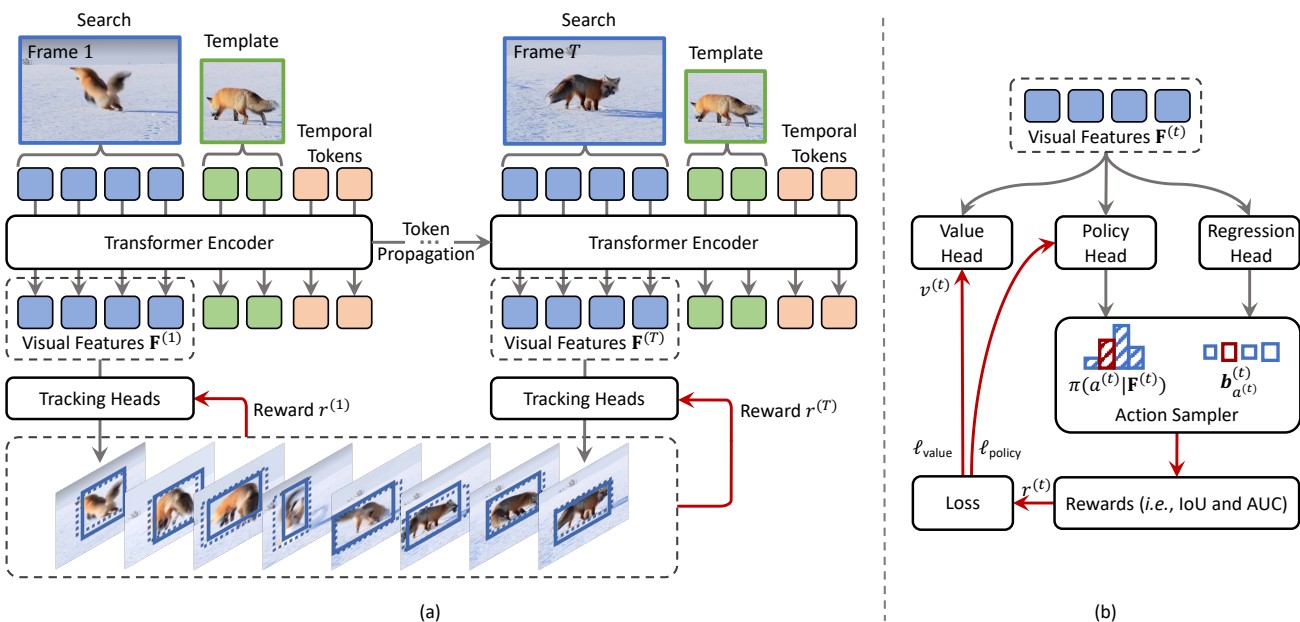

*Figure 2.* System diagram of RELO. **(a)** Policy optimization over a video clip. Only the first and last frames are shown for simplicity. For each frame, the Transformer-based encoder extracts visual features, and the tracking heads predict candidate bounding boxes. Task-oriented tracking metrics, including frame-level IoU and sequence-level AUC, are used as reward signals to optimize target localization. Layer-aligned temporal tokens are propagated across frames to provide temporal context. **(b)** Per-frame localization at frame $t$. Each spatial position on the feature map is treated as a candidate action. The policy samples one action, whose corresponding regressed box is used to compute the reward and advantage for policy optimization. Red arrows denote the flow of reward signals.

where $\beta$ balances policy learning and value regression. This policy optimization encourages reward-driven exploration: spatial locations that lead to better tracking outcomes become more likely to be selected. Importantly, because the reward need not be differentiable, it can be defined directly in terms of tracking metrics, yielding substantially better alignment between training and evaluation than prior-driven surrogate supervision.

**Regression warmup stage.** Directly optimizing the localization policy from scratch is unstable, because a sampled spatial action is useful only when the corresponding box prediction is already meaningful. We therefore begin with a regression warmup stage that learns to regress a bounding box from the visual feature at each spatial location. Concretely, for each training sample, we supervise only the single spatial location associated with the ground-truth target position using the combination of the generalized IoU (Rezatofighi et al., 2019) and $\ell_1$ losses. This stage is not intended to teach the model where to localize, but rather how to predict a bounding box from local visual features at a given location. After warmup, the components learned in this stage are frozen, which helps stabilize the subsequent policy learning stage.

**RL optimization stage.** After warmup, we optimize the localization policy on clips of $T$ consecutive search frames.

For frame $t$, the policy samples an action $a^{(t)} \sim \pi(\cdot \mid \mathbf{F}^{(t)})$, and the selected prediction is denoted by $\boldsymbol{b}_{a^{(t)}}^{(t)}$. We define the reward at frame $t$ as

$$r^{(t)} = \mathrm{IoU}\left(\boldsymbol{b}_{a^{(t)}}^{(t)}, \boldsymbol{b}_{\mathrm{gt}}^{(t)}\right) + \lambda \, \mathrm{AUC}\left(\left\{\boldsymbol{b}_{a^{(\tau)}}^{(\tau)}, \boldsymbol{b}_{\mathrm{gt}}^{(\tau)}\right\}_{\tau=1}^{T}\right), \tag{9}$$

where $\boldsymbol{b}_{\mathrm{gt}}^{(t)}$ denotes the ground-truth bounding box at frame $t$. The first term provides immediate feedback on the localization quality of the current frame, whereas the second term evaluates the sampled trajectory over the entire clip and is shared across all frames. $\lambda$ is a trade-off parameter, balancing frame-level and sequence-level supervision. In this way, the policy is encouraged not only to make accurate per-frame decisions, but also to favor action sequences that improve overall tracking performance. The resulting rewards are then used in the actor-critic objective in Eq. (5) to optimize the policy and value heads. Although training is performed on video clips, inference remains strictly frame-by-frame, thus incurring no additional test-time overhead.

### 3.3. Layer-Aligned Temporal Token Propagation

To provide temporal context across frames, RELO introduces temporal tokens that carry contextual information from frame $t - 1$ to frame $t$. A straightforward strategy, adopted in (Zheng et al., 2024), is to inject the final deep-layer temporal tokens from frame $t - 1$ into the shallow

*Table 1.* Summary of the RELO model variants, including the number of parameters, computational cost in FLOPs, and inference speed in frames per second (FPS), measured on an NVIDIA RTX 4090 GPU. The additional $+2\text{M}$ parameters correspond to the value head used during training, which is removed at test time.

| Model | #Params (M) | FLOPs (G) | Speed (FPS) |
| --- | --- | --- | --- |
| RELO-T256 | 22(+2) | 8 | 91 |
| RELO-B256 | 70(+2) | 34 | 50 |
| RELO-L256 | 247(+2) | 114 | 32 |

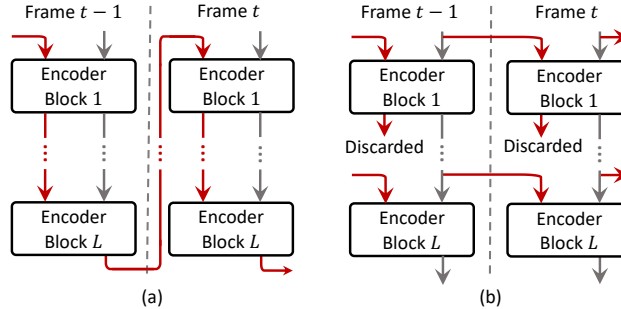

*Figure 3.* Comparison of temporal token propagation strategies. **(a)** Deep-to-shallow propagation transfers temporal tokens from the deep layers of frame $t-1$ to the shallow layers of frame $t$, introducing a semantic mismatch across layers. **(b)** Our layer-aligned propagation passes temporal tokens from each layer of frame $t-1$ to the corresponding layer of frame $t$, preserving semantic consistency during cross-frame information transfer. Red arrows denote temporal tokens propagated from the previous frame.

layers of frame $t$. However, this deep-to-shallow propagation creates a semantic mismatch: high-level representations from deep layers are forced to interact with early-layer features that primarily encode low-level appearance cues.

To address this issue, we adopt a *layer-aligned* temporal token propagation strategy. Within frame $t-1$, the temporal tokens are updated progressively from shallow to deep layers. Meanwhile, the temporal tokens produced at layer $l$ are stored and propagated only to layer $l$ of frame $t$. In this way, cross-frame information exchange is restricted to matched semantic levels, which preserves the hierarchical structure of the encoder.

More specifically, at frame $t$, we concatenate four groups of tokens as the input to the $l$-th Transformer layer: the template tokens $\mathbf{Z}_l^{(t)}$, the search tokens $\mathbf{X}_l^{(t)}$, the current-frame temporal tokens $\mathbf{T}_l^{(t)}$, and the propagated temporal tokens $\mathbf{T}_l^{(t-1)}$ from the same layer of the previous frame, which are collectively denoted by $\mathbf{H}_l^{(t)}$. The layer operation can then be written as

$$\left[\mathbf{Z}_{l+1}^{(t)},\ \mathbf{X}_{l+1}^{(t)},\ \mathbf{T}_{l+1}^{(t)},\ \mathbf{P}_{l+1}^{(t-1)}\right] = \boldsymbol{f}_l\left(\mathbf{H}_l^{(t)}\right), \qquad (10)$$

where $\boldsymbol{f}_l(\cdot)$ denotes the $l$-th Transformer layer. Once the $l$-th layer is applied, we *discard* $\mathbf{P}_{l+1}^{(t-1)}$ corresponding to the propagated tokens $\mathbf{T}_l^{(t-1)}$, and retain only $\mathbf{T}_{l+1}^{(t)}$ as the updated temporal tokens of the current frame. When entering layer $l+1$, we then concatenate these retained tokens with the layer-aligned temporal tokens propagated from frame $t-1$, yielding $\mathbf{H}_{l+1}^{(t)} = \left[\mathbf{Z}_{l+1}^{(t)},\ \mathbf{X}_{l+1}^{(t)},\ \mathbf{T}_{l+1}^{(t)},\ \mathbf{T}_{l+1}^{(t-1)}\right]$ as the input to the $(l+1)$-th Transformer layer. This design keeps the number of within-frame carried tokens unchanged across layers, preventing linear growth in token count while preserving layer-aligned cross-frame interaction.

## 4. Experiments

We evaluate RELO on seven standard visual tracking benchmarks and compare it with recent state-of-the-art trackers. We first summarize the implementation details, then report the main benchmarking results, and finally analyze the effect of the principal design choices through ablation studies.

### 4.1. Implementation Details

**Model variants.** RELO is built on a standard one-stream Transformer-based tracking framework (Ye et al., 2022). To cover different accuracy-efficiency trade-offs, we instantiate three variants of RELO with different Transformer-based encoders. Specifically, RELO-T256 uses HiViT-T (Zhang et al., 2023), RELO-B256 uses HiViT-B, and RELO-L256 uses HiViT-L, all initialized from Fast-iTPN (Tian et al., 2024). The search and template resolutions are set to $256 \times 256$ and $128 \times 128$, respectively, with a patch size of $16 \times 16$. The regression, policy, and value heads each contain five stacked Conv-BN-ReLU layers with kernel size 3 and stride 1. Spatial average pooling is applied to the per-location value map to obtain the state value used in the actor-critic objective. Unless otherwise stated, we use 32 temporal tokens; for RELO-T256, we use 4 temporal tokens for efficiency. Table 1 summarizes the model size, floating-point operations (FLOPs), and runtime of all variants.

**Training.** We train on the training splits of COCO (Lin et al., 2014), LaSOT (Fan et al., 2019), GOT-10k (Huang et al., 2019), TrackingNet (Muller et al., 2018), and Vast-Track (Peng et al., 2024). For each training sample, we draw template frames and an ordered sequence of search frames from the same video. The template and search crops are obtained by enlarging the ground-truth box by factors of 2 and 4, respectively. Data augmentation includes horizontal flipping and brightness jittering.

In the RL optimization stage, the sequence length is set to $T = 8$. RELO-B256 and RELO-L256 use two templates, whereas RELO-T256 uses a single template for improved efficiency. We set the sequence-level reward weight in Eq. (9) to $\lambda = 1$ and the policy-value loss weight to $\beta = 0.5$. All unfrozen parameters are optimized using AdamW with

*Table 2.* Comparison on large-scale benchmarks. Base-size models are listed in the upper block and large-size models in the lower block. The best and second-best entries for each metric are highlighted in **bold** and underlined, respectively.

| Method | LaSOT | | | LaSOT$_\text{ext}$ | | | TrackingNet | | | GOT-10k | | |
|---|---|---|---|---|---|---|---|---|---|---|---|---|
| | AUC | $P$ | $P_\text{Norm}$ | AUC | $P$ | $P_\text{Norm}$ | AUC | $P$ | $P_\text{Norm}$ | AO | SR$_{0.5}$ | SR$_{0.75}$ |
| OSTrack-B256 (Ye et al., 2022) | 69.1 | 75.2 | 78.7 | 47.4 | 53.3 | 57.3 | 83.1 | 82.0 | 87.8 | 71.0 | 80.4 | 68.2 |
| SwinTrack-B384 (Lin et al., 2022) | 71.3 | 76.5 | - | 49.1 | 55.6 | - | 84.0 | 82.8 | - | 72.4 | - | 67.8 |
| SeqTrack-B256 (Chen et al., 2023) | 69.9 | 76.3 | 79.7 | 49.5 | 56.3 | 60.8 | 83.3 | 82.2 | 88.3 | 74.7 | 84.7 | 71.8 |
| ARTrack-B256 (Wei et al., 2023) | 70.4 | 76.6 | 79.5 | 46.4 | 52.3 | 56.5 | 84.2 | 83.5 | 88.7 | 73.5 | 82.2 | 70.9 |
| OneTracker-B384 (Hong et al., 2024) | 70.5 | 76.5 | 79.9 | - | - | - | 83.7 | 82.7 | 88.4 | - | - | - |
| SUTrack-B224 (Chen et al., 2025) | 73.2 | 80.5 | **83.4** | 53.1 | 60.5 | 64.2 | 85.7 | 85.1 | 90.3 | 77.9 | 87.5 | 78.5 |
| ARPTrack-B256 (Liang et al., 2025) | 72.6 | 78.5 | 81.4 | 52.0 | 58.7 | 62.9 | 85.5 | 85.3 | 90.0 | 77.7 | 87.3 | 74.3 |
| RELO-B256 (Ours) | **73.3** | **80.6** | **83.4** | **54.2** | **62.2** | **65.4** | **86.4** | **86.2** | **90.7** | **80.5** | **90.5** | **81.2** |
| MixFormer-L320 (Cui et al., 2022) | 70.1 | 76.3 | 79.9 | - | - | - | 83.9 | 83.1 | 88.9 | - | - | - |
| SimTrack-L224 (Chen et al., 2022) | 70.5 | - | 79.7 | - | - | - | 83.4 | - | 87.4 | 69.8 | 78.8 | 66.0 |
| ODTrack-L384 (Zheng et al., 2024) | 74.0 | 82.3 | 84.2 | 53.9 | 61.7 | 65.4 | 86.1 | 86.7 | 91.0 | 78.2 | 87.2 | 77.3 |
| ARTrackV2-L384 (Bai et al., 2024) | 73.6 | 81.1 | 82.8 | 53.4 | 60.2 | 63.7 | 86.1 | 86.2 | 90.4 | 79.5 | 87.8 | 79.6 |
| LoRAT-L224 (Lin et al., 2024) | 74.2 | 80.9 | 83.6 | 52.8 | 60.0 | 64.7 | 85.0 | 84.4 | 89.5 | 75.7 | 84.9 | 75.0 |
| SUTrack-L224 (Chen et al., 2025) | 73.5 | 80.9 | 83.3 | 54.0 | 61.7 | 65.3 | 86.5 | 86.7 | 90.9 | 81.0 | 90.4 | 82.4 |
| ARPTrack-L384 (Liang et al., 2025) | 74.2 | 81.7 | 83.4 | 54.2 | 61.2 | 64.4 | 86.6 | 87.4 | 91.1 | 81.5 | 90.6 | 80.5 |
| RELO-L256 (Ours) | **75.1** | **83.4** | **85.1** | **57.5** | **66.7** | **69.1** | **87.3** | **88.0** | **91.6** | **81.8** | **91.1** | **83.5** |

learning rate $10^{-4}$ and weight decay $10^{-4}$. Training runs for 90 epochs with $2,500$ sampled sequences per epoch, and the learning rate is decayed by a factor of 10 after epoch 72. Additional training details are provided in Appendix C.

**Inference protocol.** At test time, RELO selects the spatial location with the highest policy score and outputs the corresponding regressed bounding box. Although RELO-B256 and RELO-L256 are trained with two templates and thus can support template updating (Yan et al., 2021; Cui et al., 2022), we disable this test-time adaptation by default to avoid overfitting to benchmark-specific characteristics. Template updating is allowed only on benchmarks where it permits a fair comparison and yields consistent performance gains, as discussed in Sec. 4.2.

## 4.2. Main Results

We evaluate RELO on seven benchmarks that probe complementary aspects of visual tracking. LaSOT (Fan et al., 2019) and LaSOT$_\text{ext}$ (Fan et al., 2021) emphasize long-horizon tracking, with LaSOT$_\text{ext}$ further testing cross-category generalization; TrackingNet (Muller et al., 2018) and GOT-10k (Huang et al., 2019) focus on large-scale short-term evaluation; TNL2K (Wang et al., 2021b), NFS (Kiani Galoogahi et al., 2017), and UAV123 (Muller et al., 2016) assess transfer beyond the main training distribution. For fairness, we compare trackers with similar model scale and input resolution. We use AUC as the primary evaluation metric. On LaSOT, LaSOT$_\text{ext}$, and TrackingNet, we additionally report precision ($P$) and normalized precision ($P_\text{Norm}$), which evaluate center-location accuracy without and with normalization by target size, respectively. On GOT-10k, we follow common practice and report average overlap (AO), *i.e.*, the mean IoU over all frames, as well as success

*Table 3.* Comparison on additional benchmarks, including TNL2K, NFS, and UAV123 in terms of AUC.

| Method | TNL2K | NFS | UAV123 |
|---|---|---|---|
| TrDiMP (Wang et al., 2021a) | - | 66.5 | 67.5 |
| TransT (Chen et al., 2021) | 50.7 | 65.7 | 69.1 |
| SimTrack (Chen et al., 2022) | 55.6 | - | 71.2 |
| OSTrack (Ye et al., 2022) | 55.9 | 66.5 | 70.7 |
| SeqTrack-L256 (Chen et al., 2023) | 56.9 | 66.9 | 69.7 |
| ARTrack-L384 (Wei et al., 2023) | 60.3 | 67.9 | 71.2 |
| LoRAT-L224 (Lin et al., 2024) | 61.1 | 66.6 | **71.9** |
| RELO-B256 (Ours) | 60.9 | 70.0 | 70.4 |
| RELO-L256 (Ours) | **63.6** | **71.3** | 71.4 |

rates (SR$_{0.5}$ and SR$_{0.75}$), corresponding to the proportions of frames whose IoU exceeds $0.5$ and $0.75$.

**Long-term tracking evaluation.** The most revealing comparison is between LaSOT and LaSOT$_\text{ext}$ in Table 2. On LaSOT, RELO-L256 achieves the best result among large-size trackers, while RELO-B256 also ranks first in the base-size group. More importantly, the advantage becomes substantially larger on LaSOT$_\text{ext}$, where both RELO variants establish clearer margins over prior methods. This contrast suggests that the main benefit of RELO is not merely stronger fitting to the training distribution, but better robustness under category shift and larger appearance variation. Existing strong baselines improve tracking mainly through stronger pretraining or broader training setups: for example, LoRAT (Lin et al., 2024) emphasizes parameter-efficient adaptation of large pretrained ViTs, whereas SUTrack (Chen et al., 2025) leverages unified training across multiple tracking settings. RELO is complementary to these directions, because it changes the localization learning objective itself by replacing handcrafted spatial priors with reward-driven action selection. The larger gain on LaSOT$_\text{ext}$ therefore

*Table 4.* Comparison with efficient trackers on large-scale benchmarks. To improve compactness, the LaSOT / LaSOT$_{ext}$ results are reported jointly in the second column, and the inference speeds on Intel Core i9-14900K CPU and NVIDIA Jetson AGX Xavier are likewise combined in the last column as "CPU / AGX."

| Method | LaSOT | TrackingNet | GOT-10k | Speed |
|---|---|---|---|---|
| FEAR-L (Borsuk et al., 2022) | 57.9 / - | - | 64.5 | - / - |
| E.T.Track (Blatter et al., 2023) | 59.1 / - | 75.0 | - | 47 / 20 |
| HiT (Kang et al., 2023) | 64.6 / - | 80.0 | 64.0 | 33 / 61 |
| MixFormerV2-S (Cui et al., 2023) | 60.6 / 43.6 | 75.8 | - | 47 / 70 |
| AsymTrack-B (Zhu et al., 2025) | 64.7 / 44.6 | 80.0 | 67.7 | 38 / 64 |
| SUTrack-T224 (Chen et al., 2025) | 69.6 / 50.2 | 82.7 | 72.7 | 23 / 34 |
| RELO-T256 (Ours) | **70.4 / 51.1** | **83.6** | **75.6** | 21 / 32 |

indicates that directly optimizing task-oriented rewards using RL is especially beneficial when center or corner priors become less reliable.

**Short-term tracking evaluation.** TrackingNet (Muller et al., 2018) and GOT-10k (Huang et al., 2019) test whether the same learning principle remains effective in the short-term setting, where template updating is commonly used and precise frame-level localization is critical. As also shown in Table 2, RELO again ranks first in both the base-size and large-size regimes on the two benchmarks. These results are noteworthy because recent strong competing methods in this regime already incorporate more elaborate temporal designs, such as video-level autoregressive pretraining in ARPTrack (Liang et al., 2025). The fact that RELO still improves over them suggests that stronger temporal representations alone do not fully resolve the localization bottleneck. Instead, learning to select spatial hypotheses with task-oriented rewards provides an additional and complementary source of gain, yielding a tracker that is strong not only on long-term benchmarks, but also on mainstream short-term evaluation protocols.

**Transfer to additional benchmarks.** The results on TNL2K, NFS, and UAV123 further support the generality of RELO (see Table 3). On TNL2K, RELO-L256 establishes the best reported AUC even without language input and benchmark-specific tuning, indicating that the learned localization policy transfers beyond the datasets used for training. On NFS, RELO achieves a clear improvement over the existing results, while on UAV123, it remains competitive with the strongest existing trackers. Taken together, these results reduce the likelihood that the gains arise from benchmark-specific training; instead, they point to a more general improvement in localization quality.

**Efficient tracking evaluation.** Finally, RELO-T256 shows that the proposed RL training paradigm is not limited to high-capacity models. Although efficient trackers typically trade accuracy for deployment speed, RELO-T256 remains ahead of existing efficient trackers on the large-

*Table 5.* Ablation results of RELO, reported in terms of AUC. The row colors indicate different groups of variants: gray denotes the baseline, green the prior-driven variants, blue the policy-optimization variants, yellow the reward-design variants, and red the temporal-modeling variants. $\Delta$ denotes the average performance change across the two benchmarks relative to the baseline.

| # | RELO variant | LaSOT | LaSOT$_{ext}$ | $\Delta$ |
|---|---|---|---|---|
| 1 | Baseline (RELO-L256) | 75.1 | 57.5 | – |
| 2 | Corner-prior training | 73.2 | 53.8 | -2.8 |
| 3 | Center-heatmap training | 73.7 | 54.0 | -2.5 |
| 4 | IoU-heatmap training | 73.7 | 54.3 | -2.3 |
| 5 | Actor-critic → PPO | 75.5 | 57.2 | +0.1 |
| 6 | Actor-critic → GRPO | 75.0 | 57.1 | -0.3 |
| 7 | w/o value-based variance reduction | 74.5 | 56.5 | -0.8 |
| 8 | w/o sequence-level AUC reward | 74.7 | 56.0 | -1.0 |
| 9 | w/o frame-level IoU reward | 75.2 | 57.1 | -0.2 |
| 10 | Reduced sequence-level learning | 74.3 | 55.5 | -1.4 |
| 11 | w/ deep-to-shallow propagation | 74.9 | 57.1 | -0.3 |

scale benchmarks while running at practical frame rates on edge-oriented hardware (see Table 4). This result is important because it shows that the benefit of RELO lies primarily in how localization is learned, rather than in model scaling alone. Consequently, the reward-driven formulation remains effective even in the low-compute regime.

**Summary.** Overall, the benchmarking results suggest a consistent picture. Prior work has improved tracking by scaling backbones, strengthening video pretraining, enriching temporal interaction, or broadening training across tasks and modalities. RELO improves a different component: the supervision used to learn where the tracker should localize. The fact that this change yields the largest gains on the most generalization-heavy benchmark, while remaining beneficial on short-term and efficient settings, supports our central claim that reward-driven localization is a stronger alternative to conventional prior-driven supervision.

### 4.3. Ablation Studies

Unless otherwise specified, all ablations use RELO-L256 as the baseline model. Results are reported in Table 5 using AUC on LaSOT and LaSOT$_{ext}$.

**Prior-driven training.** Rows 2-4 compare RELO with three conventional prior-driven localization strategies: corner-prior training (Yan et al., 2021), center-heatmap training (Ye et al., 2022), and IoU-heatmap training (Zhang et al., 2021b). For fairness, all three alternatives are implemented with the same HiViT-L backbone and aligned head architectures (see Appendix C.5 for details). All three alternatives underperform RELO, with the gap becoming more pronounced on LaSOT$_{ext}$. This result supports the central motivation of RELO: learning localization directly from task-oriented rewards is more effective than matching handcrafted spatial priors. It also suggests that surrogate priors

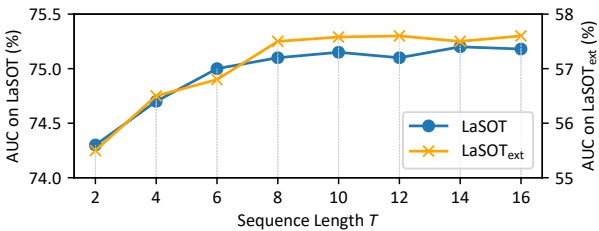

*Figure 4.* Effect of the sequence length $T$ during training. AUC on LaSOT and LaSOT$_{ext}$ improves as $T$ increases and saturates beyond $T \approx 8$, supporting the default choice of $T = 8$.

become less reliable under stronger appearance variation and distribution shift.

**Policy optimization.** Rows 5-7 evaluate different policy optimization strategies. Replacing the standard actor-critic update with PPO (Schulman et al., 2017) or GRPO (Shao et al., 2024) yields only marginal differences, indicating that the proposed localization learning does not require a more complex RL algorithm. By contrast, removing the value head causes a clear degradation. This shows that value-based variance reduction plays an important role in stabilizing policy learning, even though the overall performance gain mainly comes from the reward-driven formulation itself rather than from the choice of RL optimizer.

**Reward design.** Rows 8-9 examine the contribution of the two reward terms. Removing the sequence-level AUC reward causes a larger decrease, especially on LaSOT$_{ext}$, whereas removing the frame-level IoU reward leads to a smaller drop. This comparison suggests that sequence-level supervision is the primary factor behind the improved performance of RELO, while the frame-level IoU term provides complementary local guidance for accurate per-frame localization. The best performance is therefore obtained by combining both terms.

**Sequence-level learning.** Row 10 studies the role of sequence-level optimization by reducing the training sequence length from $T = 8$ to $T = 2$. We do not use $T = 1$, because that setting would simultaneously remove temporal token propagation and therefore confound the comparison. Shortening the sequence causes a clear degradation on both benchmarks, showing that long-horizon credit assignment is beneficial for learning the localization policy. Fig. 4 provides a more detailed view: performance improves consistently as $T$ increases before eventually saturating. This trend supports our default choice of $T = 8$ as a practical compromise between accuracy and training cost.

**Temporal token propagation.** Row 11 compares the proposed layer-aligned temporal token propagation with the deep-to-shallow propagation strategy (Zheng et al., 2024).

The aligned variant yields a consistent improvement at essentially identical computational cost. Although the gain is smaller than that of the reward design or sequence-level learning, it is nevertheless meaningful, as it shows that semantically matched temporal interaction improves feature propagation across frames.

## 5. Conclusion and Discussion

We have presented RELO, an RL-based paradigm for target localization in visual object tracking. Unlike prior-driven trackers that rely on handcrafted center or corner supervision, RELO formulates target localization as a Markov decision process over the spatial feature map and learns a localization policy directly from task-oriented rewards.

RELO is currently instantiated within a one-stream Transformer-based tracker and incorporates two practical components: a regression warmup stage that stabilizes policy learning and a layer-aligned temporal token propagation mechanism that enhances cross-frame semantic consistency with negligible inference overhead. Extensive experiments across diverse benchmarks demonstrate that RELO delivers strong and consistent performance in both long-term and short-term tracking settings. More broadly, the proposed reward-driven localization is compatible with a wide range of tracking paradigms and could be extended to SAM-based trackers (Videnovic et al., 2025), autoregressive trackers (Chen et al., 2023; Wei et al., 2023), multimodal trackers (Chen et al., 2025; Hong et al., 2024), and trackers with online adaptation (Bhat et al., 2019; Mayer et al., 2022).

Despite these encouraging results, several limitations remain. First, although IoU and AUC provide more task-relevant supervision than handcrafted spatial priors, they still do not fully capture the complexity of tracking quality in challenging scenarios such as severe occlusion, target disappearance, distractor interference, abrupt motion, and long-term recovery. An important direction for future work is therefore to design richer reward formulations that better reflect robustness, recovery ability, temporal stability, and uncertainty calibration. Second, the current formulation focuses on single-object tracking in single-camera videos. Extending reward-driven localization to multi-object and multi-camera settings is a promising next step, but it will likely require new designs for joint action modeling, cross-object competition, identity preservation, and cross-view consistency. Third, RL in RELO is currently used only for target localization, whereas a practical tracker involves a broader set of sequential decisions. Future work could extend RL to the full tracking pipeline, including template updating, memory management, search-region adaptation, re-detection, and other system-level control decisions, so that the tracker can learn coordinated policies for both *where* to localize and *how* to maintain reliable tracking over time.

## Acknowledgments

This work was supported in part by the Hong Kong ITC Innovation and Technology Fund (9440379 and 9440390) and Liaoning Provincial Science and Technology Joint Program Project (2024011188-JH2/1026).

## Impact Statement

This work advances localization learning for visual object tracking and may benefit applications in robotics, autonomous systems, and human-computer interaction. However, like other tracking methods, it could be misused for privacy-invasive surveillance or unauthorized monitoring, and errors or biases may create risks in safety-critical settings. We therefore encourage deployment with consent, oversight, and careful robustness analysis.

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

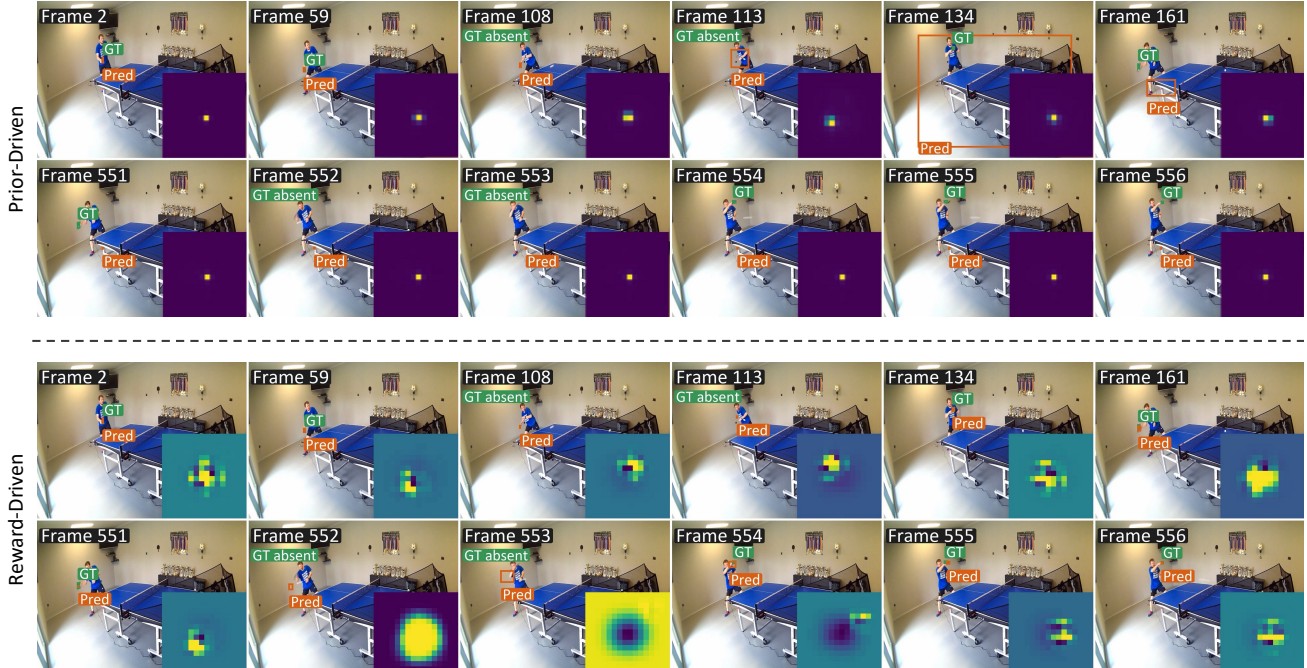

*Figure 5.* Qualitative comparison between the reward-driven RELO and the prior-driven tracker in a failure-and-recovery scenario. Frame indices are shown in the upper-left corner. For each frame, the lower-right inset visualizes the localization score map over the search region using the `viridis` colormap, where yellow indicates high confidence and dark blue indicates low confidence. Green and orange boxes denote the ground-truth (GT) and predicted bounding boxes, respectively. Frames without ground-truth boxes correspond to cases where the target is absent or too ambiguous to annotate. Best viewed in color with zoom.

## A. Qualitative Results

RELO exhibits a distinctive *recovery* behavior after target loss, which emerges naturally from reward-driven localization without handcrafted spatial priors. As shown in Fig. 5, RELO tracks the paddle reliably and maintains a concentrated score map around the target. After frame 551, however, the target becomes highly ambiguous and briefly disappears. At frame 552, RELO expands its search beyond the central region but does not yet recover the target. At frame 553, the tracker shifts probability mass toward the outer search region, where the target reappears in the upper-right corner. Over frames 554-556, the prediction progressively re-aligns with the target and fully recovers.

The center-prior baseline behaves differently. Because its localization branch is trained with an explicit center prior, the score map remains concentrated near the presumed target location even after failure. As a result, it explores the search region less effectively and is less likely to recover once the target leaves the expected area.

Fig. 6 provides additional qualitative comparisons across challenging scenarios, including occlusion, out-of-view motion, deformation, distractors, and background clutter. Across these cases, RELO generally yields tighter and more stable localization results.

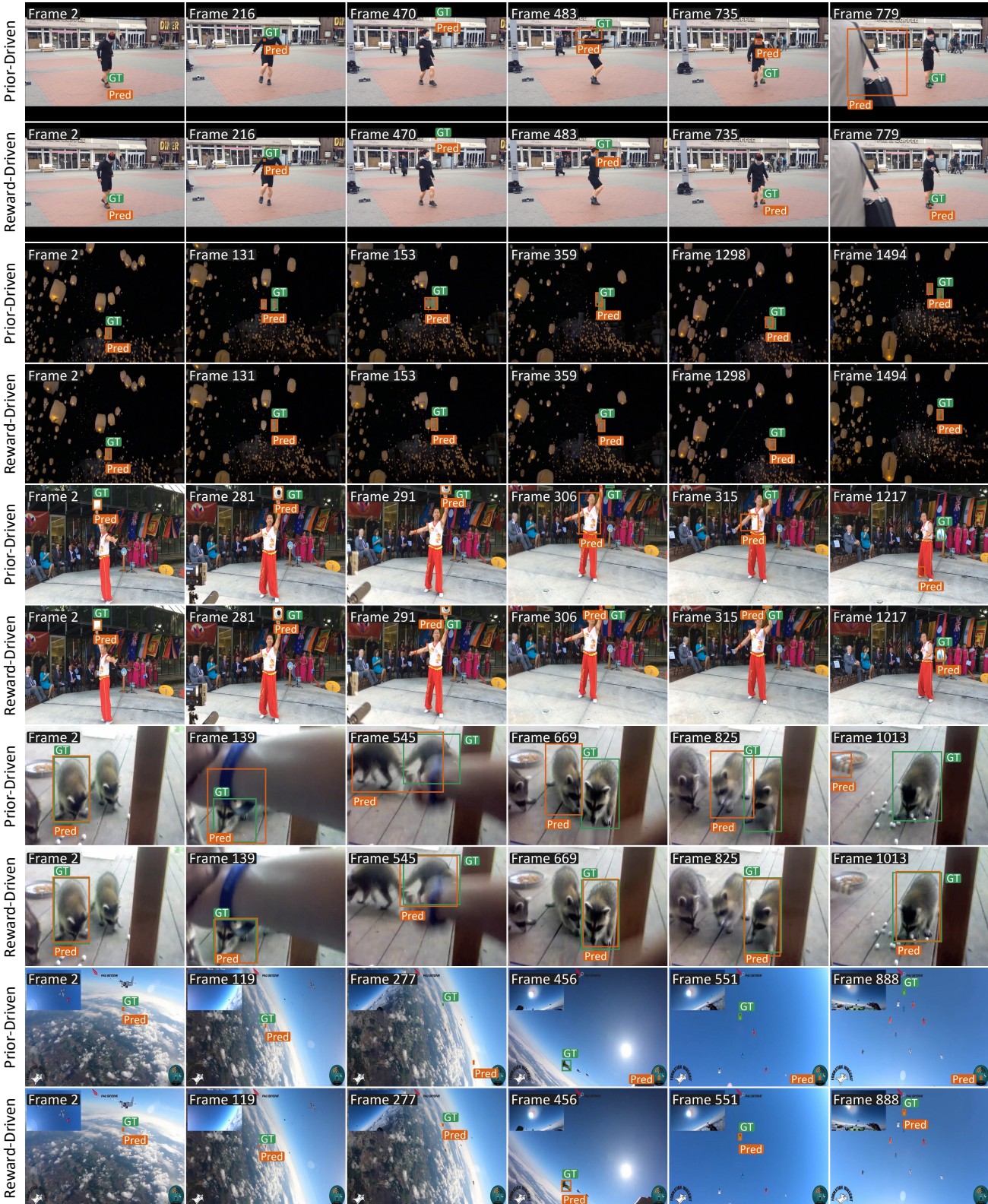

*Figure 6.* Additional qualitative comparisons between the reward-driven RELO and the prior-driven tracker across challenging scenarios.

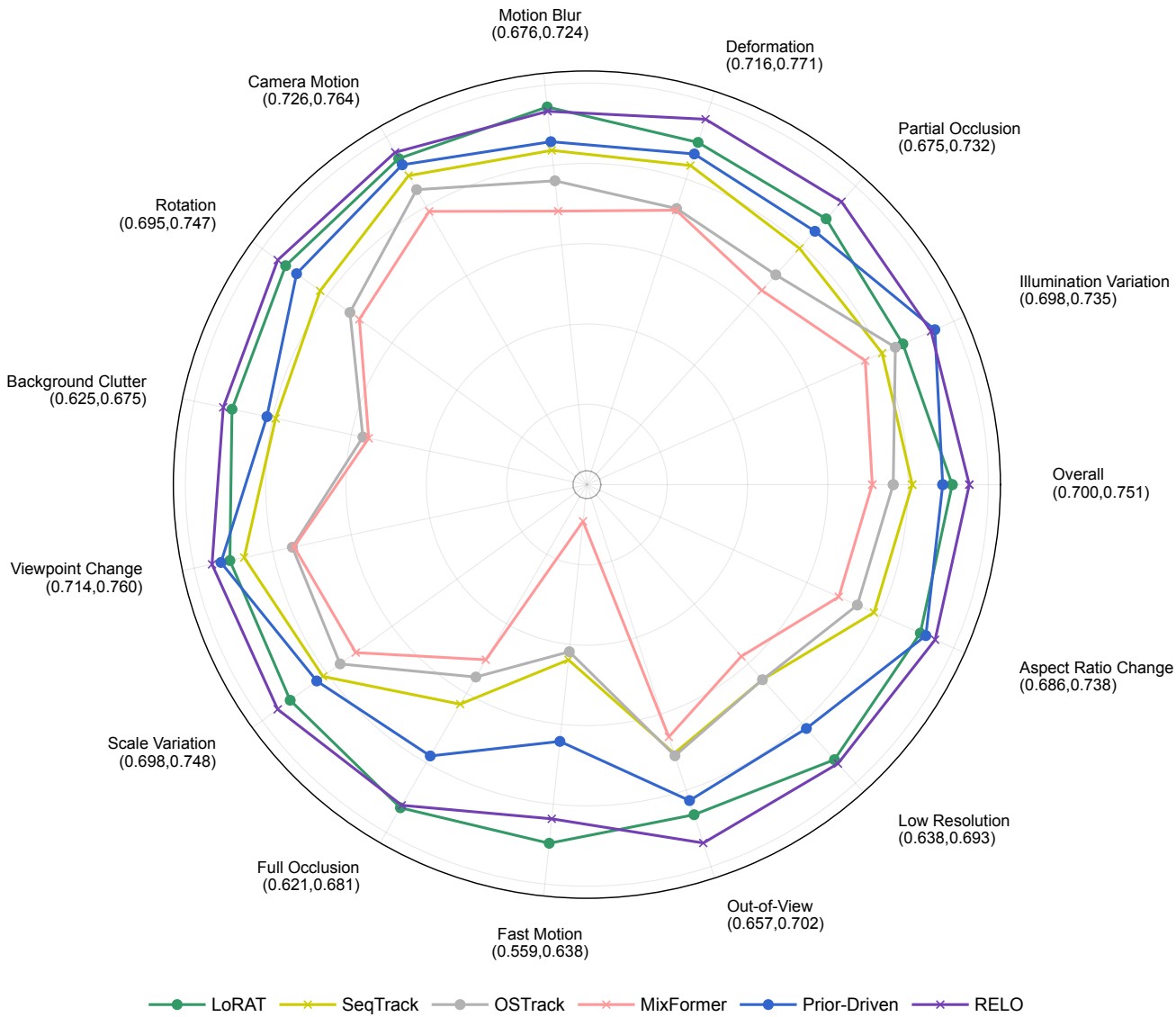

*Figure 7.* Attribute-wise AUC comparison on LaSOT. For each attribute, the two numbers in parentheses indicate the minimum and maximum AUC values achieved by the compared trackers, respectively.

## B. More Analysis

This section presents supplementary analysis that complements the results in the main text.

**Attribute-wise performance.** Fig. 7 reports attribute-wise AUC on LaSOT. We include the center-prior-driven tracker trained under settings aligned with RELO, to facilitate a controlled comparison. RELO is competitive across most attributes and improves most clearly on *fast motion*, *out-of-view*, *full/partial occlusion*, *deformation*, *scale variation*, *low resolution*, and *background clutter*. These are typically the cases in which rigid center-biased localization is most restrictive. By learning a reward-driven policy over spatial positions, RELO explores the search region more flexibly and remains more robust when the target moves unpredictably or deviates substantially from the search center.

**Alternative training strategies.** Table 6 evaluates several variants of the default training pipeline. Unfreezing the Transformer-based encoder and/or regression head during the RL stage (in Rows 2-3) slightly reduces performance. We therefore keep these components fixed after regression warmup so that policy learning operates in a more stable environment. Training the regression branch and localization policy jointly from scratch (in Row 4) causes a substantial performance drop. Early in training, noisy box predictions make the reward landscape unstable, which induces the policy to collapse onto a

*Table 6.* Additional ablation results of RELO on LaSOT and LaSOT$_{ext}$.

| # | RELO variant | LaSOT | LaSOT$_{ext}$ | $\Delta$ |
|---|---|---|---|---|
| 1 | Baseline (RELO-L256) | 75.1 | 57.5 | – |
| 2 | Unfreezing Transformer-based encoder + regression head during RL | 74.6 | 56.8 | -0.6 |
| 3 | Unfreezing regression head only during RL | 74.8 | 56.9 | -0.5 |
| 4 | w/o regression warmup | 70.2 | 51.1 | -5.7 |
| 5 | Initializing policy with a prior-driven localizer | 73.9 | 54.4 | -2.2 |

*Table 7.* Effect of the policy-value loss weight $\beta$ in the actor-critic objective.

| $\beta$ | LaSOT | LaSOT$_{ext}$ | $\Delta$ |
|---|---|---|---|
| 0.25 | 75.1 | 57.3 | -0.1 |
| 0.50 | 75.1 | 57.5 | – |
| 1.00 | 75.0 | 57.1 | -0.3 |
| 2.00 | 74.7 | 56.6 | -0.7 |

small set of locations before meaningful exploration can occur. This result confirms that regression warmup is important for stable policy optimization. Initializing the localization policy with parameters from a center-prior localization head (in Row 5) also degrades performance. The inherited center bias narrows exploration and limits learning from task-oriented rewards. The best results are obtained when the localization policy is learned from scratch, without handcrafted priors.

**Effect of the policy-value loss weight.** We evaluate $\beta \in \{0.25, 0.5, 1.0, 2.0\}$ in the actor-critic objective, whose results are shown in Table 7. RELO is robust across a broad range of values. Settings between $0.25$ and $1.0$ perform nearly identically, with $\beta = 0.5$ yielding the best overall result. A larger value ($\beta = 2.0$) reduces accuracy, likely because optimization becomes overly focused on value prediction and weakens the effective policy-gradient signal. We therefore use $\beta = 0.5$ by default.

**Effect of the number of temporal tokens.** We also study the number of temporal tokens. Table 8 shows that removing temporal tokens entirely still yields competitive performance, but adding a small number consistently improves accuracy. Performance remains stable across a broad range (*e.g.*, 8-64 tokens). Our default choice of 32 temporal tokens achieves the best overall accuracy with negligible computational overhead.

## C. More Implementation Details

This section provides additional implementation details that complement the descriptions in the main text.

### C.1. Total Training Images

The total number of search images used in the regression warmup stage is

$$100,000 \times 90 \times 2 = 18,000,000.$$

The total number of search images used in the RL optimization stage is

$$2,500 \times 90 \times 8 = 1,800,000.$$

Overall, the total training volume is comparable to that of existing Transformer-based tracking pipelines such as OSTrack (Ye et al., 2022) and SUTrack (Chen et al., 2025), resulting in a similar overall training cost for models of comparable size.

### C.2. Data Sampling and Augmentation

As described in Sec. 4.1 of the main text, we train on five datasets: COCO (Lin et al., 2014), LaSOT (Fan et al., 2019), GOT-10k (Huang et al., 2019), TrackingNet (Muller et al., 2018), and VastTrack (Peng et al., 2024). Training samples are drawn uniformly across datasets to avoid imbalance.

*Table 8.* Effect of the number of temporal tokens.

| #Temporal tokens | LaSOT | LaSOT$_{\text{ext}}$ | $\Delta$ |
|:---:|:---:|:---:|:---:|
| 0 | 74.7 | 56.8 | -0.6 |
| 8 | 75.1 | 57.3 | -0.1 |
| 16 | 75.0 | 57.4 | -0.1 |
| 32 | 75.1 | 57.5 | – |
| 64 | 75.2 | 57.4 | +0.0 |

For video datasets (LaSOT, GOT-10k, TrackingNet, and VastTrack), we first sample a video and then sample frames from it, with a maximum temporal stride of $400$ frames to preserve reasonable temporal consistency. For the image-based dataset COCO, we follow common practice and treat each image as a static video by duplicating it across the required number of frames. This makes COCO compatible with the video-based sampling pipeline.

To preserve temporal consistency during augmentation, operations such as horizontal flipping are applied jointly to all frames in a video sequence rather than independently to each frame.

### C.3. Regression Warmup

During warmup, the sequence length is set to $T{=}2$, since sequence-level learning is not yet involved. We use a learning rate of $10^{-5}$ for the Transformer-based encoder and $10^{-4}$ for the remaining modules, with weight decay $10^{-4}$. Training runs for 90 epochs with 100,000 sampled sequences per epoch, and the learning rate is decayed by a factor of 10 after epoch 72.

### C.4. Reward Computation

For completeness, we provide the mathematical definitions of the IoU and AUC scores used in our reward computation.

**Intersection over union (IoU).**  For frame $t$, let the predicted bounding box be $\boldsymbol{b}^{(t)}$ and the ground-truth box be $\boldsymbol{b}_{\text{gt}}^{(t)}$. Denote by $\text{area}(\cdot)$ the area of a box and by $\text{Int}(\cdot, \cdot)$ the intersection region of two boxes. The frame-level IoU score is

$$\text{IoU}\left(\boldsymbol{b}^{(t)}, \boldsymbol{b}_{\text{gt}}^{(t)}\right) = \frac{\text{area}\left(\text{Int}\left(\boldsymbol{b}^{(t)}, \boldsymbol{b}_{\text{gt}}^{(t)}\right)\right)}{\text{area}\left(\boldsymbol{b}^{(t)}\right) + \text{area}\left(\boldsymbol{b}_{\text{gt}}^{(t)}\right) - \text{area}\left(\text{Int}\left(\boldsymbol{b}^{(t)}, \boldsymbol{b}_{\text{gt}}^{(t)}\right)\right)}. \tag{11}$$

**Success curve.**  The success rate at threshold $\rho \in [0, 1]$ is defined as

$$S(\rho) = \frac{1}{T} \sum_{t=1}^{T} \mathbb{I}\left[\text{IoU}^{(t)} > \rho\right], \tag{12}$$

where $\mathbb{I}[\cdot]$ is the indicator function. In practice, we use the standard discrete threshold set $\rho_k \in \{0.00, 0.05, \dots, 1.00\}$.

**Area under the success curve (AUC).**  The AUC score is defined as the integral of the success curve:

$$\text{AUC} = \int_0^1 S(\rho) \, d\rho. \tag{13}$$

Using the discrete threshold set $\{\rho_k\}_{k=1}^K$, the AUC score is approximated by

$$\text{AUC} \approx \frac{1}{K} \sum_{k=1}^{K} S(\rho_k). \tag{14}$$

### C.5. Ablation Study Details

This subsection provides additional details for the ablations in the main text.

**Prior-driven training variants.** For the three *prior-driven training* variants in the main paper (Sec. 4.3, Table 5, Rows 2-4), we consider the following standard implementations.

*Corner-prior training* follows the standard corner-head design of STARK (Yan et al., 2021): the model predicts top-left and bottom-right probability maps, and the final box is decoded by taking expectations over the spatial distributions. *Center-heatmap training* follows that in OSTrack (Ye et al., 2022), where a Gaussian-smoothed heatmap centered on the target is used as supervision. *IoU-heatmap training* constructs a supervisory heatmap by computing, at each spatial location, the IoU between the regressed box and the ground-truth box. All variants use the same HiViT-L backbone and aligned head architecture as RELO to ensure a controlled comparison.

**Alternative policy optimizers.** In the main text, we evaluate PPO (Schulman et al., 2017) and GRPO (Shao et al., 2024) as alternatives to the standard actor-critic update used in RELO. For PPO, we use the clipped surrogate objective. To avoid conflict with the reward notation $r^{(t)}$, we denote the importance ratio by

$$\varrho^{(t)} = \frac{\pi(a^{(t)} \mid \mathbf{F}^{(t)})}{\pi_{\mathrm{old}}(a^{(t)} \mid \mathbf{F}^{(t)})},$$

where $\pi_{\mathrm{old}}$ denotes the behavior policy used to sample the actions, *i.e.*, a frozen snapshot of the current policy from the previous update period. It is held fixed when computing the PPO importance ratio and is refreshed every four optimization steps in our implementation. The PPO loss is then defined as

$$\ell_{\mathrm{PPO}} = -\frac{1}{T} \sum_{t=1}^{T} \min\left( \varrho^{(t)} A^{(t)}, \mathrm{clip}\left(\varrho^{(t)}, 1 - \epsilon_{\mathrm{ppo}}, 1 + \epsilon_{\mathrm{ppo}}\right) A^{(t)} \right), \tag{15}$$

where $\epsilon_{\mathrm{ppo}}$ is set to $0.2$. We do not use any regularization, since the policy is trained from scratch and does not need to be anchored to a pretrained reference model.

For GRPO, we follow the standard group-based formulation, whose key difference from PPO is the use of group-normalized advantages. For each video sequence, we sample $G$ predicted trajectories ($G = 8$ in our implementation). For trajectory $g \in \{1, \ldots, G\}$, we compute per-frame rewards $\{r_g^{(t)}\}_{t=1}^{T}$ for actions $\{a_g^{(t)}\}_{t=1}^{T}$ using the same reward design as RELO. For each frame $t$, the corresponding reward group is $\{r_1^{(t)}, r_2^{(t)}, \ldots, r_G^{(t)}\}$. The GRPO advantage is obtained by normalizing each reward within the group:

$$A_{\mathrm{grp},g}^{(t)} = \frac{r_g^{(t)} - \mathrm{mean}\left( \left\{ r_{g'}^{(t)} \right\}_{g'=1}^{G} \right)}{\mathrm{std}\left( \left\{ r_{g'}^{(t)} \right\}_{g'=1}^{G} \right) + \epsilon}, \tag{16}$$

where $\epsilon > 0$ is a small constant added for numerical stability to avoid potential division by zero. Analogous to PPO, we define the per-trajectory importance ratio

$$\varrho_g^{(t)} = \frac{\pi(a_g^{(t)} \mid \mathbf{F}^{(t)})}{\pi_{\mathrm{old}}(a_g^{(t)} \mid \mathbf{F}^{(t)})},$$

so that the GRPO loss becomes

$$\ell_{\mathrm{GRPO}} = -\frac{1}{G} \sum_{g=1}^{G} \frac{1}{T} \sum_{t=1}^{T} \min\left( \varrho_g^{(t)} A_{\mathrm{grp},g}^{(t)}, \mathrm{clip}\left(\varrho_g^{(t)}, 1 - \epsilon_{\mathrm{ppo}}, 1 + \epsilon_{\mathrm{ppo}}\right) A_{\mathrm{grp},g}^{(t)} \right). \tag{17}$$

