# OpenReview forum: "RELO: Reinforcement Learning to Localize for Visual Object Tracking"
_ICML.cc/2026/Conference — ICML 2026 regular_

### Official Review · Reviewer_641H · 2026-02-24

**Soundness:** 3
**Presentation:** 3
**Significance:** 3
**Originality:** 3
**Overall Recommendation:** 4
**Confidence:** 3

**Summary:**

the paper propose RELO, a reinforcement-learning tracking framework that formulates target localization as a decision-making problem within the Transformerbased tracking paradigm.

**Compliance With Llm Reviewing Policy:**

Affirmed.

**Final Justification:**

The rebuttal addressed my concerns. I will keep my positive rating.

**Key Questions For Authors:**

- Can sequence-level prediction improve inference? Is it possible for the sequence-level prediction used in inference time to improve performance (such as majority vote)?
- How sensitive is performance to reward weighting (λ₁, λ₂)?

**Limitations:**

yes

**Strengths And Weaknesses:**

Strength
- The idea of introducing reinforcement learning into dense localization for tracking is conceptually interesting and seems to be novel. Instead of treating localization as supervised heatmap regression, the paper formulates it as a policy over spatial locations, which is a meaningful shift in perspective.
- The method achieves strong and consistent improvements across multiple benchmarks (LaSOT, LaSOText, TrackingNet, GOT-10k),
 especially in generalization-heavy settings like LaSOText.
- paper is well written and easy to follow.
- The paper perform extensive results on both cpu and gpu for efficiency evaluation, which provides evidence on the practical relevance of the proposed method.

Weakness
- Missing comparison with foundation-model-based trackers. lack of comparison with foundation model like sam2 [1] and sam2-based methods such as samurai [2], sam2long [3]
- is it possible for the sequence-level prediction used in inference time to improve performance (such as majority vote)?
- Applicability beyond one-stream Transformer trackers is unclear. The method is tightly coupled with a specific tracking paradigm. Its generalization to other frameworks such as autoregressive trackers is not demonstrated and tested in the paper.

---

[1] Ravi, Nikhila, et al. "Sam 2: Segment anything in images and videos." arXiv preprint arXiv:2408.00714 (2024).

[2] Yang, Cheng-Yeng, et al. "SAMURAI: Motion-Aware Memory for Training-Free Visual Object Tracking with SAM 2." IEEE Transactions on Image Processing (2026).

[3] Ding, Shuangrui, et al. "Sam2long: Enhancing sam 2 for long video segmentation with a training-free memory tree." Proceedings of the IEEE/CVF International Conference on Computer Vision. 2025.

---

> ### Author Rebuttal · Authors · 2026-03-28
>
> Thank you for your positive assessment of the novelty, empirical performance, clarity, and practical efficiency of our paper. We also appreciate the constructive suggestions. We address your concerns below.
>
> **Sequence-level prediction at inference time.** Thank you for this insightful question. We agree that sequence-level prediction at inference time is an interesting direction and could potentially further improve robustness. RELO incorporates sequence-level supervision during training through the AUC reward, which encourages good sequence-level tracking behavior. At inference time, we use simple frame-wise decoding to preserve the causal and real-time nature of online tracking. A true majority-vote scheme over future frames would break the online setting, while a causal alternative such as short-window temporal ensembling over past predictions is possible. Such temporal aggregation may improve stability in some cases, but it can also introduce latency. We have not systematically evaluated these variants in the current submission, so we prefer not to overclaim. Nevertheless, we agree this is an interesting direction and will mention it as future work.
>
> **Sensitivity to reward weighting.** Thank you for your detailed comment. To further address it, we additionally tested different reward weights. These results suggest that RELO is not highly sensitive to the reward weights within a reasonable range. As long as both the AUC and IoU terms are included, the performance remains consistently competitive. We will add this clarification in the revision.
>
> | Setting | λ1 (IoU) | λ2 (AUC) | LaSOT | LaSOText |
> |---|---:|---:|---:|---:|
> | \#1 | 1.5 | 0.5 | 75.2 | 57.3 |
> | \#2 (Baseline) | 1.0 | 1.0 | 75.1 | 57.5 |
> | \#3 | 0.5 | 1.5 | 75.0 | 57.5 |
>
> **Comparison with SAM-based trackers.**   Thank you for your helpful comment. We summarize representative SAM-based trackers in the table below. Overall, RELO shows competitive performance relative to SAM-based trackers. RELO also has an efficiency advantage on our machine with an NVIDIA RTX 4090 GPU under an online tracking setting: SAM-based methods run at around 15 FPS, whereas RELO-T256 / B256 / L256 run at 91 / 50 / 32 FPS, respectively.
>
> | Method | Input Resolution | LaSOT | LaSOText | GOT-10k |
> |---|---:|---:|---:|---:|
> | SAM2-L | 1024 | 70.0 | 56.9 | 80.7 |
> | SAM2Long-L | 1024 | 73.9 | 60.9 | 81.1 |
> | SAMURAI-B | 1024 | 70.7 | 57.5 | 79.6 |
> | SAMURAI-L | 1024 | 74.2 | 61.0 | 81.7 |
> | **RELO-B256** | 256 | 73.3 | 54.2 | 80.5 |
> | **RELO-L256** | 256 | 75.1 | 57.5 | 81.8 |
>
> We do not view one-stream Transformer-based trackers like RELO and foundation-model-based trackers like SAM2 as cases where one clearly dominates the other. Instead, we believe they focus on different regimes and offer different strengths. RELO performs search-region tracking with box-level prediction, using lower input resolution. This makes it more suitable for accurate bounding-box prediction and latency-sensitive scenarios with lower training and inference cost. By contrast, SAM-based methods are attractive when full-image context, high-resolution mask prediction, and promptable tracking are more important than latency. We will add this discussion in the revision.
>
> More broadly, we also see it as an interesting future direction to combine RELO with SAM-based trackers. In particular, the RELO formulation may help improve localization accuracy, memory-update decisions, or mask selection in such foundation-model-based tracking frameworks. Thank you again for this helpful suggestion.
>
> **Applicability beyond one-stream Transformer trackers.** We appreciate this thoughtful comment. The current submission focuses on one-stream dense-prediction tracking paradigm, as it provides a clean and widely adopted setting. More broadly, RELO mainly requires a candidate action space and a task-level reward, so extension to other tracker families is feasible. (1) For Siamese-style dense-prediction trackers, actions can be defined over candidate locations or boxes. (2) For autoregressive trackers, actions can be reformulated as prediction tokens or token sequences. (3) For SAM-based trackers, actions can be defined over candidate masks or prompts with mask-based IoU/AUC rewards. (4) The same idea may also apply to other decision points in tracking, such as memory update, where the action space is explicit.  Extending RELO beyond one-stream trackers is a promising future direction, and we hope the current work can serve as a useful starting point.
>
> Thank you again for your valuable review.

---

> > ### Author Rebuttal · Reviewer_641H · 2026-04-01
> >
> > The rebuttal addressed my concerns. I will maintain my positive rating.

---

### Official Review · Reviewer_TdQx · 2026-03-12

**Soundness:** 4
**Presentation:** 3
**Significance:** 4
**Originality:** 3
**Overall Recommendation:** 5
**Confidence:** 5

**Summary:**

This paper proposes RELO, a reinforcement-learning-based framework for Transformer-based visual object trackers. Instead of training the localization module with prior-based heatmap supervision, RELO treats each spatial position on the search feature map as a discrete action, learns a localization policy over positions, and optimizes that policy using rewards based on task-level performance. The method first performs a regression warm-up stage to learn box prediction, then freezes the regression-related components and trains policy/value heads with a policy-gradient objective. The paper also introduces a lightweight layer-wise temporal token alignment mechanism. Experiments on multiple tracking benchmarks show strong results, especially on LaSOText, while keeping inference real-time and without extra test-time overhead.

**Compliance With Llm Reviewing Policy:**

Affirmed.

**Final Justification:**

My concerns are addressed, so I maintain my positive rating.

**Key Questions For Authors:**

The paper is novel and technically compelling, and I am overall positive about it. However, I would like to see the author's responses to my comments in Weaknesses 1 and 2.

**Limitations:**

yes

**Strengths And Weaknesses:**

Strengths:
1. The paper is strong in motivation. It identifies a genuine mismatch in Transformer-based tracking: localization is trained with handcrafted spatial priors or heatmap-like surrogate supervision, while evaluation is based on task-level criteria such as IoU and AUC. This motivation is compelling and clearly articulated.
2. The proposed method is novel and well aligned with the motivation. Both in the warm-up stage and in the RL stage, the method indeed moves away from handcrafted spatial priors, which is different with previous SFT-based or RL-based methods. Moreover, RL is well integrated into the dense-prediction tracking pipeline rather than acting as an external module. This is clean and practical for modern end-to-end trackers.
3. The experiments are convincing. The method shows promising results on tracking benchmarks, especially on LaSOText. The inference speed is real-time. The paper also includes controlled ablations on prior-driven supervision schemes, policy optimization variants, rewards, and temporal modeling choices.
4. The presentation is generally good. The paper is well structured, the main figures are easy to follow, and the qualitative results make the intended behavior of the method intuitive.
5. Overall, I think this is a meaningful contribution for the tracking community, since it offers a practical way to replace handcrafted localization supervision with task-aligned training while retaining a standard inference pipeline. It may also help bridge tracking or dense prediction research with the growing interest in RL–based alignment paradigms that have become prominent in the LLM/MLLM literature.
Weaknesses:
1. The current validation is mainly centered on one-stream Transformer-based trackers. Although this is already an important and relevant setting, it would be interesting to better understand how transferable the proposed localization-learning paradigm is to other tracking families.
2. Beyond directly matching tracking metrics in the reward design, further exploration of reward signals could be interesting, for example incorporating MLLM-based feedback rewards.
3. The color block in Table 5 may be difficult to read on some displays, which can affect readability.
4. The visualization of localization behavior in Appendix Figure 5 provides important evidence for the method’s motivation and characteristics. Moving it into the main paper would strengthen the presentation.

---

> ### Author Rebuttal · Authors · 2026-03-27
>
> Thank you for your positive assessment and for recognizing the motivation, novelty, convincing experiments, and practical significance of RELO. We are encouraged that you view RELO as a meaningful contribution to the tracking community. We address your comments below.
>
> **Extension beyond one-stream trackers.**
> We appreciate your insightful comment. The current submission intentionally focuses on one-stream dense-prediction tracking paradigm, as it provides a clean and widely adopted setting. More broadly, RELO mainly requires a candidate action space and a task-level reward, so extension to other tracker families is feasible. (1) For Siamese-style dense-prediction trackers, actions can be defined over candidate locations or boxes. (2) For autoregressive trackers, actions can be reformulated as prediction tokens or token sequences. (3) For SAM-based trackers, actions can be defined over candidate masks or prompts with mask-based IoU/AUC rewards. (4) The same idea may also apply to other decision points in tracking, such as memory update, where the action space is explicit.
> Extending RELO beyond one-stream trackers is a promising future direction, and we look forward to exploring it in future work.
>
> **Rewards.**
> Thank you for your insightful suggestion. Our current choice of IoU and AUC is deliberate: it directly aligns training with widely used tracking metrics while keeping the study controlled and focused on the central question of whether handcrafted prior-based localization supervision can be replaced by task-level optimization. At the same time, RELO does not require the reward to be differentiable, and is therefore naturally compatible with richer feedback signals, including MLLM-based semantic or identity-consistency rewards. We did not explore such rewards in the current submission in order to keep the paper focused and to avoid confounding the core contribution with additional reward-engineering choices. Nevertheless, we agree that this is a promising direction, and we hope RELO can serve as a solid baseline for exploring richer reward designs for tracking.
>
> **Presentation issues.**
> Thank you for your detailed comments. We agree that the color coding in Table 5 can be improved for readability, and that moving the localization-behavior visualization from the appendix to the main paper would strengthen the presentation. We will revise both accordingly.
>
> Thank you again for your valuable review.

---

> > ### Author Rebuttal · Reviewer_TdQx · 2026-04-01
> >
> > The author has answered my questions from 6 to 9, and I have no other concerns, so I maintain the original rating.

---

### Official Review · Reviewer_kwVK · 2026-03-13

**Soundness:** 3
**Presentation:** 3
**Significance:** 3
**Originality:** 3
**Overall Recommendation:** 4
**Confidence:** 3

**Summary:**

This paper focuses on the visual object tracking task.  To localize the objects, previous methods often use localization heads to predict the spatial priors.  This paper formulates the tracking task in a different way, proposing to align the training objective with evaluation criteria using reinforcement learning. Experimental results show that the proposed method achieves state-of-the-art tracking results on multiple benchmarks, such as LaSOT, demonstrating the effectiveness of the proposed method.

**Compliance With Llm Reviewing Policy:**

Affirmed.

**Final Justification:**

Thanks to the authors for the rebuttal, which solved my concerns. I will keep my positive rating.

**Key Questions For Authors:**

Please refer to the weakness part.

**Limitations:**

This paper discusses the limitations, but does not include the potential negative societal impact.

**Strengths And Weaknesses:**

**Strength:**

* Different from previous methods trained to fit the spatial priors, this paper formulates visual object tracking into a reinforcement learning problem optimized with task-level rewards, which is interesting and insightful.
* This paper points out the feature consistency in previous deep-to-shallow propagation, and proposes a new  layer-wise temporal
token alignment method, which is novel.
* This work provides comprehensive experiments to support the effectiveness of the proposed method, such as comparing with diverse baseline methods on multiple benchmarks. This paper also presents an extensive ablation study to analyze the contributions of the design elements in the proposed method.
* The paper is well presented and easy to follow.

**Weakness:**
* This paper shows visualization results in diverse scenarios (e.g., occlusion) in the appendix. While I understand the limit of space in the main paper, it would be recommended to include more visualization and discussions in the main paper to help the readers have a better understanding of the proposed method.
* Table 5 shows that the IoU award provides a limited influence on the final result compared to the AUC award.  Is there any insight into this result?
* As mentioned in the limitations, this paper focuses on one-stream Transformer-based tracking architectures. Is it possible to directly use the method in other architectures, e.g., SAM-based or autoregressive trackers?  Further discussion of different architectures is recommended.

---

> ### Author Rebuttal · Authors · 2026-03-27
>
> Thank you for your detailed review and positive feedback. We appreciate your recognition of RELO’s RL-based tracking formulation, the novelty of the temporal token alignment, the comprehensiveness of the experiments and ablations, and the clarity of the paper’s presentation. We also value your helpful suggestions for further strengthening the manuscript.
>
> **Visualization results.**
> Thank you for your helpful suggestion. We agree that moving representative visualizations and discussion from the appendix to the main paper would help readers better understand the method. We will streamline the presentation accordingly in the revision.
>
> **IoU and AUC rewards.**
> Thank you for this insightful question. Our intuition is that the sequence-level AUC reward is more directly aligned with the final video-level tracking performance. In contrast, instantaneous IoU only reflects the quality of the current frame. In addition, since AUC is computed from frame-wise overlaps over the whole sequence, it already captures IoU-related information to some extent. Therefore, the AUC reward plays the dominant role, while the IoU reward mainly serves as a frame-level shaping signal that provides additional emphasis on current-frame localization quality.
>
> **Extension beyond one-stream trackers.**
> Thank you for your constructive suggestion. The current submission intentionally focuses on one-stream dense-prediction tracking paradigm, as it provides a clean and widely adopted setting. More broadly, RELO mainly requires a candidate action space and a task-level reward, so extension to other tracker families is feasible. (1) For Siamese-style dense-prediction trackers, actions can be defined over candidate locations or boxes. (2) For autoregressive trackers, actions can be reformulated as prediction tokens or token sequences. (3) For SAM-based trackers, actions can be defined over candidate masks or prompts with mask-based IoU/AUC rewards. (4) The same idea may also apply to other decision points in tracking, such as memory update, where the action space is explicit.
> We agree that further discussion of extending RELO beyond one-stream tracking would strengthen the paper, and we will add this discussion in the revision.
>
> **Potential negative societal impact.**
> Thank you for raising this point. While our work focuses on generic tracking on public benchmarks, tracking methods of this kind could be misused for privacy-invasive surveillance or unauthorized monitoring, and tracking failures or biases may cause harm in safety-critical settings. We will add a brief discussion of these risks in the revised version.
>
> Thank you again for your valuable review.

---

> > ### Author Rebuttal · Reviewer_kwVK · 2026-04-03
> >
> > Thanks to the authors for the rebuttal, which solved my concerns. I will keep my positive rating.

---

### Official Review · Reviewer_DrQ2 · 2026-03-17

**Soundness:** 3
**Presentation:** 3
**Significance:** 3
**Originality:** 3
**Overall Recommendation:** 3
**Confidence:** 5

**Summary:**

This paper proposes RELO, an RL-based framework for one-stream Transformer visual trackers, solving the training-evaluation misalignment caused by handcrafted spatial prior heatmaps.  RELO performs sequence-level reinforcement learning to optimize localization behavior using both instantaneous IoU and sequence-level AUC rewards. Experiments show RELO achieves SOTA performance with real-time GPU inference, and ablations validate core designs.

**Compliance With Llm Reviewing Policy:**

Affirmed.

**Key Questions For Authors:**

1.Why do PPO/GRPO not outperform vanilla policy gradient, and has hyperparameter tuning for these algorithms been tested?
2.What is the temporal token alignment’s computational overhead vs. ODTrack, and why only modest gains on challenging temporal attributes?
3.Why choose a linear IoU/AUC reward, and have weighted/non-linear formulations been tested for tracking-specific challenges?
4.Can RELO’s RL formulation adapt to non-one-stream trackers (e.g., SAM-based), and what modifications are needed?
5.Could you supplement the quantitative metrics of training cost such as single-epoch training time, GPU memory peak, and total computational cost, and further conduct a comparative analysis with SOTA trackers to quantify the training cost-performance ratio of RELO?

**Limitations:**

No.

**Strengths And Weaknesses:**

Strengths:
1.The authors conduct fair and extensive comparisons against state-of-the-art methods, covering diverse scenarios to ensure the generalizability of the results.
2.The authors validate the necessity of the regression warm-up strategy through detailed ablation studies, demonstrating its critical role in stabilizing training.

Weaknesses:
1.Algorithm Selection: There is no substantive discussion on why sophisticated RL optimizers like PPO or GRPO underperform compared to vanilla policy gradient. This omission leaves the choice of optimizer unjustified.
2.Efficiency Analysis: The proposed temporal token alignment yields only marginal improvements, yet the paper fails to report the corresponding computational costs. A rigorous efficiency-accuracy trade-off analysis is missing.
3.Reward Design: The reward function is restricted to a naive linear combination of IoU and AUC. The authors provide no ablation studies on alternative reward formulations, raising concerns about the method's robustness in complex, real-world tracking scenarios."

---

> ### Author Rebuttal · Authors · 2026-03-27
>
> We appreciate your insights and address your concerns below.
>
> **1. RL optimizer.**
> Thank you for your constructive comment. Our ablation aims to test whether RELO requires a sophisticated optimizer, rather than to claim that advanced RL optimizers are ineffective in general. Empirically, under standard implementations, PPO and GRPO are comparable to vanilla policy gradient. A plausible reason is that the RL stage in RELO is already relatively stable: warm-up provides a stable regression environment, the action space is moderate, the value head reduces variance, and the reward is informative. Under this regime, clipping or group normalization may offer limited extra benefit. We will clarify in the revision that PPO/GRPO are tested under standard settings rather than exhaustive tuning. The main takeaway is that RELO does not critically rely on a sophisticated RL optimizer. We also view more advanced RL optimizers for tracking as an interesting direction, and hope RELO can serve as a solid baseline.
>
> **2. Efficiency of temporal token alignment.**
> Thank you for the detailed comment. The proposed alignment changes *how* temporal tokens are propagated across layers (Fig. 3). The aligned propagation introduces no additional overhead. In the ablation, we isolate the propagation strategy while keeping the rest unchanged. The gain is modest, which is expected because the comparison is between two temporal-token propagation variants, rather than between temporal modeling and no temporal modeling. Thus, the improvement comes at essentially no extra cost, making the design favorable in practice.
>
> **3. Reward design.**
> Thank you for your valuable comment.  We use a linear combination of IoU and AUC rewards to ensure training aligned with the tracking evaluation criteria, allowing us to isolate the core contribution of RELO: replacing handcrafted prior-based localization supervision with direct task-level optimization. Tab. 5 of the main paper studies the two reward terms. To further address this concern, we additionally tested different weights:
>
> | Setting | λ1 (IoU) | λ2 (AUC) | LaSOT | LaSOText |
> |---|---:|---:|---:|---:|
> | \#1 | 1.5 | 0.5 | 75.2 | 57.3 |
> | \#2 (Baseline) | 1.0 | 1.0 | 75.1 | 57.5 |
> | \#3 | 0.5 | 1.5 | 75.0 | 57.5 |
>
> These results suggest that RELO is not highly sensitive to the reward weights within a reasonable range. We have not exhaustively explored more complex rewards in the current submission, but we agree that they are an important direction for future work. We hope RELO can provide a useful starting point for such future research.
>
> **4. Extension beyond one-stream trackers.**
> We appreciate your insightful question. The current submission intentionally focuses on one-stream dense-prediction tracking paradigm, as it provides a clean and widely adopted setting. More broadly, RELO mainly requires a candidate action space and a task-level reward, so extension to other tracker families is feasible. (1) For Siamese-style dense-prediction trackers, actions can be defined over candidate locations or boxes. (2) For autoregressive trackers, actions can be reformulated as prediction tokens or token sequences. (3) For SAM-based trackers, actions can be defined over candidate masks or prompts with mask-based IoU/AUC rewards. (4) The same idea may also apply to other decision points in tracking, such as memory update, where the action space is explicit.
> Extending RELO beyond one-stream trackers is a promising future direction, and we look forward to exploring it in future work.
>
> **5. Training cost.**
> Following your suggestion, we measured RELO and several representative trackers on the same 4×A40 server and report unified training-cost statistics. Since training schedules differ across trackers, we consider these statistics more informative than single-epoch time. Under this unified setup, RELO is not the lowest-cost method in raw GPU-hours: SUTrack is somewhat more training-efficient. However, RELO achieves stronger final accuracy with only moderate extra cost, while remaining substantially cheaper than SeqTrack and ODTrack.
>
> | Method | Peak Mem | Train Time | GPUh | LaSOT | LaSOText | TrackingNet | GOT-10k |
> |---|---:|---:|---:|---:|---:|---:|---:|
> | SeqTrack-B256 | 29.9 GB | 84.0 h | 336.0 | 69.9 | 49.5 | 83.3 | 74.7 |
> | ODTrack-B | 41.8 GB | 73.8 h | 295.1 | 73.2 | 52.4 | 85.1 | 77.0 |
> | SUTrack-B224 | 45.3 GB | 37.8 h | 151.3 | 73.2 | 53.1 | 85.7 | 77.9 |
> | **RELO-B256** | 45.3 GB | 39.1 h | 156.5 | 73.3 | 54.2 | 86.4 | 80.5 |
> | SeqTrack-L256 | 42.0 GB | 257.2 h | 1028.8 | 72.1 | 50.5 | 85.0 | 74.5 |
> | ODTrack-L | 32.2 GB | 240.4 h | 961.5 | 74.0 | 53.9 | 86.1 | 78.2 |
> | SUTrack-L224 | 45.4 GB | 112.1 h | 448.5 | 73.5 | 54.0 | 86.5 | 81.0 |
> | **RELO-L256** | 38.1 GB | 116.7 h | 467.0 | 75.1 | 57.5 | 87.3 | 81.8 |
>
> **Note:** Peak memory also depends on the per-GPU batch size, so a large model can show lower peak memory than a base model.
>
> Thank you again for your valuable review.

---

### Decision · Program_Chairs · 2026-04-30

**Decision:**

Accept (regular)

**Comment:**

RELO presents a conceptually novel and technically sound approach to visual tracking by replacing heuristic heatmap supervision with direct reinforcement learning of evaluation metrics. The paper is well‑written, the experiments are extensive and reproducible, and the rebuttal resolved all initial concerns with concrete additional data. The positive assessments from two reviewers and the lack of sustained objection from the third reviewer, combined with the strong empirical results and clear contribution, place this work well above the acceptance threshold for ICML 2026.